# Application of Chitosan-Lignosulfonate Composite Coating Film in Grape Preservation and Study on the Difference in Metabolites in Fruit Wine

**Boran Hu [1,\*], Lan Lin [1], Yujie Fang [1], Min Zhou [1] and Xiaoyan Zhou [1,2,\*]**

[1] School of Food Science and Engineering, Yangzhou University, Yangzhou 225127, China; linlan_yang2023@163.com (L.L.); fang.yujie431@mail.kyutech.jp (Y.F.); lovelylovelymia@163.com (M.Z.)
[2] Key Laboratory of Chinese Cuisine Intangible Cultural Heritage Technology Inheritance, Ministry of Culture and Tourism, Yangzhou 225127, China
\* Correspondence: huboran@yzu.edu.cn (B.H.); yzuxyz@163.com (X.Z.)

**Abstract:** In order to solve the global problem of fruit rotting due to microbial infection and water loss after harvest, which leads to a large amount of food waste, this experiment uses degradable biological composite coating to prolong the preservation period of grapes. Chitosan (CH) and Lignosulfonate (LS) were used as Bio-based film materials, CH films, 1% CH/LS films and 2% CH/LS biomass composite films were synthesized by the classical casting method and applied to grape preservation packaging. Its preservation effect was tested by grape spoilage rate, water loss rate, hardness, soluble solids, titratable acid, and compared with plastic packaging material PE film. At the same time, $^1$H NMR technology combined with pattern recognition analysis (PCA) and partial least squares discriminant analysis (PLS-DA) was used to determine the nuclear magnetic resonance (NMR) of Cabernet Sauvignon, Chardonnay and Italian Riesling wines from the eastern foothills of Helan Mountain to explore the differences in metabolites of wine. The results of preservation showed that the grapes quality of CH films and 2% CH/LS coating package is better than the control group, the decay rates decreased from 37.71% to 21.63% and 18.36%, respectively, the hardness increased from 6.83 to 10.4 and 12.78 and the soluble solids increased from 2.1 in the control group to 3.0 and 3.2. In terms of wine metabolites, there are similar types of metabolites between cabernet Sauvignon dry red wine and Chardonnay and Italian Riesling dry white wine, but there are significant differences in content. The study found that 2% CH/LS coating package could not only reduce the spoilage rate of grapes, inhibit the consumption of soluble solids and titratable acids, but also effectively extend the shelf life of grapes by 6 days.

**Keywords:** grapes; food preservation; chitosan; biomass composite coating; wine; metabolites; proton nuclear magnetic resonance technology

## 1. Introduction

Grapes are fresh and juicy, bright in color, sweet and sour and rich in nutritional value [1]. However, because they are harvested in the high-temperature season, the fruit stem is a typical respiratory climacteric type [2], the physiological metabolism is vigorous after harvest, and the fruit tissue is crisp and tender. During storage, it is very prone to water loss, stem withering, browning and easily infected by bacteria and rot, resulting in a short storage cycle, which seriously affects the appearance quality and commodity value of fresh grapes, causing huge economic losses and food waste. At present, the preservation measures of grapes at home and abroad mainly use sulfur dioxide ($SO_2$) preservatives, coating preservation, low-temperature refrigeration and other technologies to prolong the storage period of grapes [3]. Although $SO_2$ has a good effect on maintaining the quality of grapes, it is easy to cause certain bleaching damage to the fruit, affect the original flavor, and the $SO_2$ remaining in the fruit will be harmful to human health [4]. Packaged food preservation is one of the effective

solutions. However, the packaging material commonly used in the preservation process is plastic; for example, PE, PS, or PET [5]. Chitosan is a kind of natural edible macromolecular polysaccharide that is non-toxic, has no smell, with good film forming characteristics [6,7]. After coating treatment, a colorless and transparent biological film is formed on the grape surface, which can prevent water loss and reduce the rate of weight loss in the fruit. It can also effectively prevent microbial invasion, reduce the respiratory intensity of the fruit [8–10] to a certain extent, and reduce the decay of the fruit, thus prolonging its storage time [11]. However, the coating effect of single chitosan is not stable [12], the drying time is long, the water permeability is high, and the coating toughness is poor. Sodium lignosulfonate is an anionic polymer surfactant extracted from papermaking waste liquor [13]. Because of its similar structure to lignin, it is often used as a bio-based polymer material [14]. LS has a good function but cannot form a film by itself. We tried adding LS to CH to make films suitable for fruit preservation, because the use of bio-based materials for packaging can not only be used as cheap packaging for fresh fruits and vegetables, but also reduce the pollution of plastic packaging to the environment. It achieves the requirements of highly efficient anticorrosion, low carbon and environmental protection.

Wine is fresh grapes or grape juice as raw materials, through all or part of the fermentation brewing, containing a certain degree of alcohol fermented wine [15]. It is not only a nutritious beverage, but moderate drinking can prevent various chronic diseases and enhance human health [16–18]. Fresh grapes are the key raw materials for wine brewing, and the style of wine is closely related to the variety of wine grapes, the climate of the place of origin, soil conditions and the distinctive brewing processes, making the wines produced in different producing areas have different flavors [19]. The eastern foot of Helan Mountain in Ningxia is located in the "golden" area of grape planting at 30 to 40 latitudes from north to south. The superior geographical location, unique landform features, suitable soil and climate conditions mean that wine grapes in the eastern foot of Helan Mountain in Ningxia fully possess excellent brewing characteristics.

With the rapid development of the world economy and the improvement of people's quality of life, the wine industry has developed rapidly and has become an increasingly popular product for consumers [20], but the consumer's ability to distinguish is limited. Usually, physical and chemical indexes and sensory evaluation are used to identify the quality of wine grapes and wine, but it is difficult to reflect various metabolites in wine grapes and wine that affect their quality and are beneficial to human health through these indicators [21]. In studies on wine metabolites, a "metabolic fingerprint" is generally provided based on [1]H-NMR technology [22,23]. Gregory et al. [24] used NMR to analyze and study metabolites in French red wine in the region of Bordeaux, so as to better distinguish red wine of Bordeaux from other red wines produced in French wine regions. In chemometrics, pattern recognition is the main method used to solve the attribution problem and marker search in complex systems, among which PLS-DA is the most important pattern recognition method applied in metabonomics. It is widely used in plant, drug metabolomics and food source determination and classification research [25]. Godelmann R. et al. [26] used nuclear magnetic resonance technology combined with multivariate statistics and principal component analysis to analyze and study the target compounds and non-target compounds of wine metabolites, and the results showed that the accuracy of variety identification reached 95%, the accuracy of age identification reached 97% and 96%, respectively, and the accuracy of origin identification reached 89%. All these results showed that [1]H-NMR combined with multivariate analysis was an extraordinary effective method to identify different wine varieties and region.

This paper investigated the preparation method of separate 2% chitosan film (CH), chitosan—1% sodium lignosulfonate film (1% CH/LS) and chitosan—2% sodium lignosulfonate film (2% CH/LS), and the film forming properties of the three films were studied. The preservation effect of 2% CH/LS film on grape berry was then studied further and compared with PE film (Control) and Chitosan film (CH). The differences in metabolites in Cabernet Sauvignon, Chardonnay and Italian Riesling wines at the eastern foot of Helan

Mountain in Ningxia were also studied to determine the biomarkers that contribute to the differences, so as to provide the cornerstone for wine quality control, variety identification and protection, Additionally, we also provide a scientific theoretical basis for consumers to choose high quality wine.

## 2. Materials and Methods

### 2.1. Reagents and Equipment

Chitosan($C_6H_{11}NO_4$, Average molecular weight (MW): 150 kDa; Degree of deacetylation $\geq$ 90%) were purchased from Shanghai Lanji Biological Technology Co., Ltd. (Shanghai, China); Sodium-Lignosulfonate (Content $\geq$ 98%) were purchased from Hefei BOSF Biotechnology Co., Ltd. (Hefei, China); Ascorbic acid (Vc) was obtained from Sinopharm Group Chemical Reagent Co., Ltd. (Shanghai, China); Oxalic acid, Sodium oxalate were produced by Su Yi Chemical Reagent Co., Ltd., Shanghai, China; DSS was produced by Qingdao Tenglong Microwave Technology Co., Ltd. (Qingdao, China); Heavy water ($D_2O$ deuterium degree > 99.9%) is from Tenglong Weibo Technology Co., Ltd., Qingdao, China.

SNL315SV-230 Freeze dryer was produced by Termo Co., Ltd., Waltham, MA, USA; AVANCE 600 Nuclear magnetic resonance spectrometer was supplied by Bruker Co., Ltd., Karlsruhe, German; TG16A-WS Desktop high-speed centrifuge was produced by Lu Xiangyi Centrifuge Instrument Co., Ltd., Shanghai, China; BS-224 Electronic balance was produced by Eppendorf Co., Ltd., Hamburg, German; ULT178-6-V49 Ultra low temperature freezer was supplied by Revco Co., Ltd., Waltham, MA, USA; RE52-4 Rotary evaporator was produced by Huxi Analysis Instrument Co., Ltd., Shanghai, China; XW-80A Swirl mixer was supplied by Huxi Analysis Instrument Co., Ltd., Shanghai, China.

### 2.2. Methods

#### 2.2.1. Preparation Method CH/LS Bio-Composite Film

LS powder (1 g) was dissolved in 50 mL of distilled water at 5000 rpm for 15 min. CH powder (1 g) was dissolved in 50 mL of distilled water at 5000 rpm for 15 min at 25 °C. Then, VC (1 g) and chitosan was added at 5000 rpm for 15 min by stirring continuously. Finally, the CH/LS film-forming solutions were degassed to remove air bubbles [27]. The 20 mL CH/LS film-forming solution was cast on a 15 cm in diameter flat glass Petri dish (B-SLPYM90, BKMAM Biological Co., Ltd., Changsha, China) and dried in an oven at 35 °C for 24 h until the surface of CH/LS film remained certain firm and viscous. Finally, the CH/LS film was stored in a small desiccator at 50% relative humidity for 20 min at 25 °C. Peel the CH/LS film from the Petri dish for analysis. The prepared films were, respectively added with 0% LS + 2% CH, 1% LS + 2% CH, 2% LS + 2% CH, so they were named as CH/LS, 1% CH/LS, 2% CH/LS, respectively [28].

#### 2.2.2. Determination Method of CH/LS Film on the Preservation Effect of Grape Berry

Seasonal fresh grapes (Cabernet Sauvignon variety, from Ningxia, China) were tested in research packaging material experiments. Grapes of similar size and quality were tightly wrapped with the CH/LS (2%) and CH (2%) films and incubated at a constant temperature of 18 °C in an incubator with relative humidity of 68% to observe the surface changes, compared with those wrapped in polyethylene (20 cm × 20 cm) [29].

#### 2.2.3. Determination of Decay Rates

Decay rates refer to the ratio of the weight of rotten fruit to the total weight of treated fruit [30]. The main observations related to whether the appearance of the fruit is full, whether it has edible value, and whether it is corrupted bacterial infection and deterioration. The decay rates of grapes were then calculated according to Equation (1):

Decay rates (%) = number of rotten fruits (N)/total number of fruits (N0) × 100%     (1)

### 2.2.4. Hardness

The hardness was checked using a fruit hardness tester (GY-1, JC Group Co., Ltd., Qingdao, China). Before measurement, the hardness tester was placed perpendicular to the surface of the grapes (without breaking the skin), the indenter was pressed into the fruit evenly, and the reading indicated when the pointer stops moving was the hardness value of the fruit [31].

### 2.2.5. Weight Loss Rate

The weight loss rate was measured using the weighing method [32]. The weight of the fruit on the 0th to the 16th day was measured and the average value was taken to calculate the weight loss rate.

The formula for calculating the weight loss rate content is as Equation (2):

$$\text{Weight loss rate (\%)} = \text{fruit weight (m) g/fruit original weight (m0) g} \times 100\% \text{ in a single measurement.} \tag{2}$$

### 2.2.6. Soluble Solids

Soluble solids refer to sugars, acids, vitamins and minerals that are soluble in water in fruit juice. After the grapes were broken and homogenized, soluble solids were determined with a PAL-1 hand-held sugar calorimeter [33]; the unit was %. Each group was tested 3 times in parallel.

### 2.2.7. Titratable Acid Content

The content of titratable acid was determined by acid-base titration. First, 0.5 g of a grapes sample was weighed, ground thoroughly, and then transferred to a 100 mL volumetric flask, where distilled water was added to the mark and thixotropic. Then it was filtered with filter paper and we accurately drew 20 mL of the filtrate into a 100 mL conical flask and added 2 drops of 1% phenolphthalein indicator. Using calibrated 0.01 m mol/L sodium hydroxide to titrate to pH 8.0 as the end point, we recorded the sodium hydroxide consumption. We repeated this 3 times to obtain the average value [34]. The formula for calculating the titratable acid content is as follows:

$$\text{Total acidity (\%)} = V \times C \times N \times \text{Conversion factor} \times 100/(W \times V1) \tag{3}$$

V: The total volume of sample dilution (mL); V1: the volume of the sampling solution during titration (mL);
C: The number of milliliters of sodium hydroxide standard solution consumed;
N: The molar concentration of sodium hydroxide standard solution;
W: Sample weight (g);
Conversion factor: Tartaric acid—0.075.

### 2.3. Determination of Metabolites in Wine

The Wine Sample

The grapes and wines used in this experiment were provided by Ningxia (Yinchuan) Helan Mountain Grape Wine Co., Ltd. (Yinchuan, China) and all the single varieties of wine in 2020 were made by standard brewing techniques; the same brewer's yeast (Lalvin CY 3079) was added during the brewing process for fermentation. The physical and chemical indicators of product quality are in compliance with the national standard GB15037-2006 [35]. The samples were stored at −4 °C for later use.

### 2.4. NMR Spectroscopic Analysis

2.4.1. Pre-Treatment of Wine Samples

After freeze-drying, the sample was treated with buffer solution: First, 10 mL of wine, was centrifuged at 4000 rpm for 10 min at −4 °C; 3 mL supernatant was then placed into a 20 mL lyophilized bottle and frozen overnight at −70 °C. After, it was then frozen in a freeze dryer for 48 h. Next, 400 μL of 0.2 mol/L oxalate buffer was added with pH = 4

prepared by $D_2O$, 140 μL of $D_2O$ and 60 μL of 0.75% DSS, mixed well and centrifuged at 13,000 rpm for 10 min. Finally, 500 μL of the supernatant was taken and loaded into a 5 mm nuclear magnet tube; NMR experiments were carried out.

### 2.4.2. NMR Experimental Data Collection

[1]H-NMR spectra of wine samples were collected by AVANCE 600 NMR spectrometer. The NMR experiment was set to a constant temperature of 298 K, the [1]H-NMR operating frequency was 600.23 MHz, and the spectral width was 7183.9 Hz. The Noesygppr1d sequence was used to suppress the water peak signal, and all the samples were scanned 256 times.

### 2.5. Statistical Analysis

Statistical analysis was performed by Microsoft Excel 2011 and one-way analysis of variance (ANOVA) using SPSS Statistics 23.0 software. The confidence level was 95% and $p < 0.05$ was considered statistically significant. The experimental data shown in all of the results were repeated at least three times.

### NMR Spectral Data Processing

After the sampling was completed, Fourier transform was performed, and the phase adjustment and baseline correction of the spectrum were performed; the spectral peaks were assigned according to the chemical shift. Using AMIX software, the spectrum was integrated into the chemical shift interval δ 0.5–10.0 ppm in the 0.005 ppm integration section; −0.5~0.5 ppm, 1.74~1.84 ppm and 2.90~2.95 ppm DSS peaks, 1.18~1.22 ppm and 3.57–3.72 ppm residual ethanol peaks, 4.8~4.96 ppm residual water peaks were not integrated. After the NMR data was normalized, it was imported into SIMCA-P 12.0 software for pattern recognition analysis.

## 3. Results and Discussion

### 3.1. The Effect of CH/LS Film on the Preservation of Grapes

### 3.1.1. Determination of Decay Rates

Grapes still have vigorous life activities after picking, and a series of physiological and biochemical changes occur in the fruit tissue after picking, resulting in fruit rot. The effects of different preservation treatments on the decay rates of grapes are shown in Figure 1. It can be seen that the control, CH and 2% CH/LS film packaging had no significant effect on the decay rate of grapes at the early stage of storage. After 2% CH/LS film, CH, and PE (control) packaging, the decay rates of grapes were 18.36%, 21.63%, 31.71%, respectively. However, with the extension of storage time, the decay rates of grapes demonstrated an increase, and the difference in preservation effect of the three groups was also completely different. After CH and 2% CH/LS coating film packaging, the decay rates of grapes decreased by 10.08% and 13.35%, respectively, compared with the control group. The grapes packaged in control film showed severe spoilage after 10 days of storage at 18 °C, while the grapes packaged with chitosan film and 2% CH/LS film suffered severe spoilage on the 14th and 16th days, respectively.

### 3.1.2. Hardness

Fruit hardness is an important indicator for measuring fruit quality and reflecting fruit ripening and senescence [36]. The effects of different treatments on the firmness of grapes are shown in Figure 2. The hardness of the grapes in each treatment group showed a downward trend, and the fruit hardness of the control group decreased most significantly: the hardness decreased by 13.05, during the 16th day of the experimental period. The fruit hardness of the experimental group decreased slowly and was significantly different from the control group ($p < 0.05$). Among them, the 2% CH/LS treatment group had the slowest decline: during the 16-day experimental period, the hardness decreased by only 6. The addition of lignosulfonate improved the coating effect of chitosan to a certain extent [37].The

plump appearance of the fruit was maintained; the hardness of grapes treated with CH or 2% CH/LS composite film coating was higher than that of the control group, and the effect of maintaining the hardness was obvious, and there was no significant difference between the two ($p > 0.05$), indicating that the composite film solution is beneficial for grapes to maintain fruit firmness, delay fruit softening and rot, and prolong the storage period.

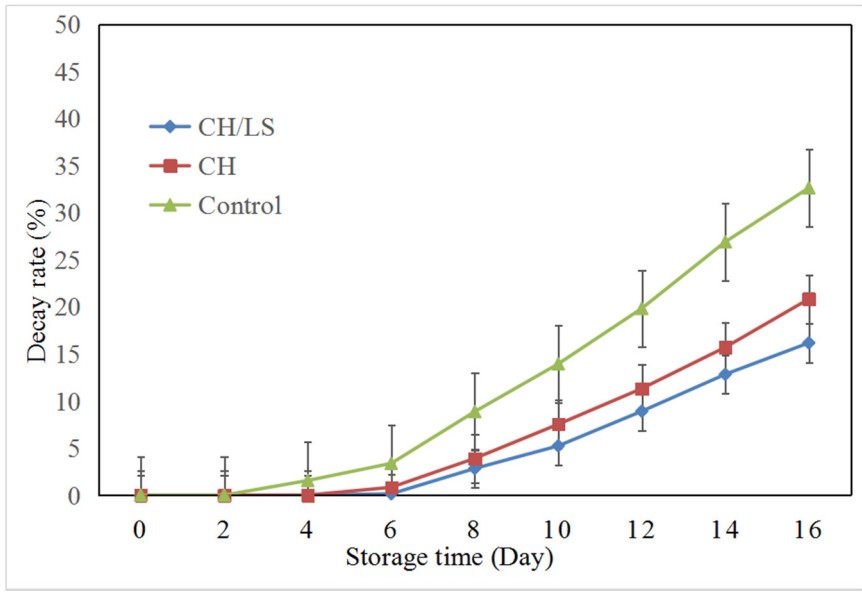

**Figure 1.** Decay rates of grapes during preservation with packaging films.

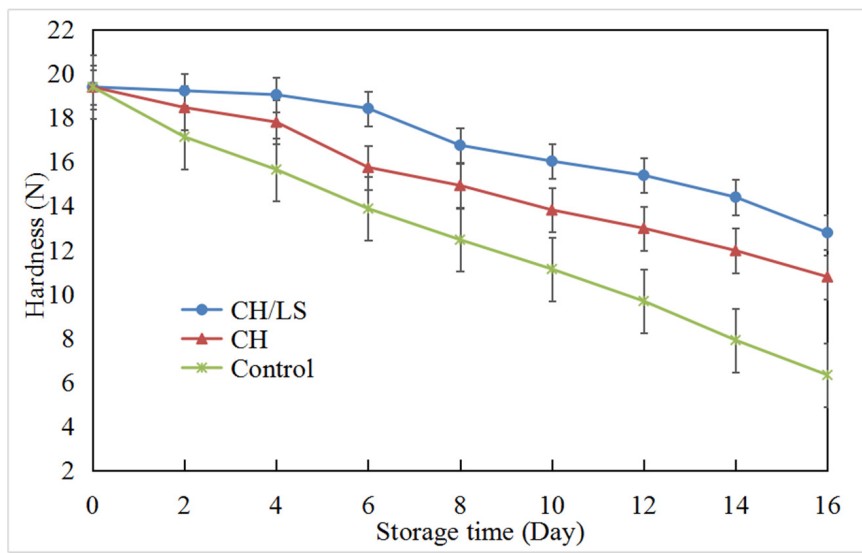

**Figure 2.** Hardness of grapes during preservation with different packaging films.

### 3.1.3. Weight Loss Rate

It is generally believed that respiration and transpiration are the main reasons for fruit weight loss. The effect of different coating treatments on the weight loss rate of grapes is shown in Figure 3. With the prolongation of storage time, the weight loss rates of grapes in CH, 2% CH/LS and control groups reached 25.49%, 21.37% and 23.24%, respectively. The weight loss rate of the CH group was higher than the control. The reason for this may be that CH film packaging has a large water vapor transmission coefficient, leading to a high water loss rate [38]. Compared with CH, 2% CH/LS film packaging can more effectively reduce the rapid loss of moisture and organic matter in grapes. The reason may be that

LS is hygroscopic, forming a barrier between the fruit epidermis and the surrounding environment to prevent gas exchange, inhibited the respiration and transpiration of grapes, reduced the water loss of grapes, and the nutrient decomposition was slower, so as to reduce the weight loss rate [39]. Although the weight loss rate of different treatment groups increased, 2% CH/LS was significantly lower than that of the rest of the two groups, the water loss of grapes was reduced, and the nutrient decomposition was slower than that of the control group.

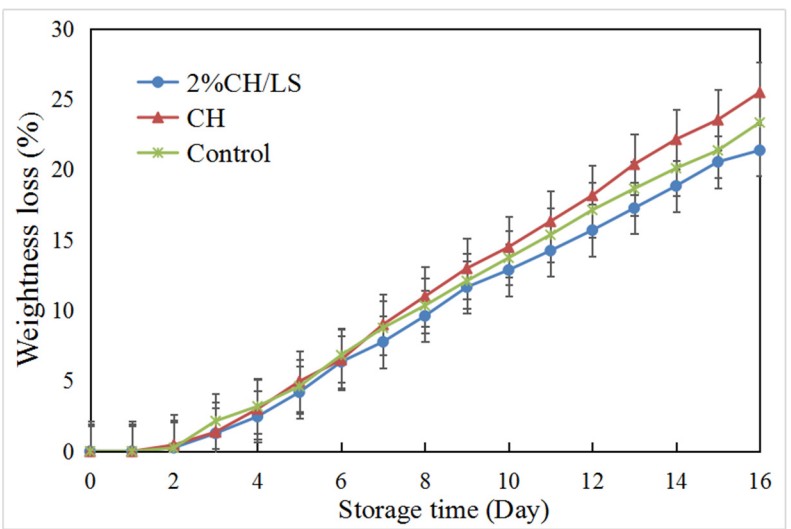

**Figure 3.** Weight loss rate of grapes during preservation with different packaging films.

3.1.4. Soluble Solids

The effects of different coating treatments on the soluble solids content in grapes are shown in Figure 4. As grapes respiration proceeds, there is no external source of nutrients, resulting in a decrease in the soluble solids content of grapes [40]. The changing trend of soluble solid content among different components is the same. The soluble solid content in the fruit of the control group decreased from 5.92% on 0 d to 2.14% on the 16th day. At the end of storage for 16 days, the 2% CH/LS coating film packaging group and the CH coating film packaging group had a better preservation effect, and the soluble solid content was about 3.05% and 3.28%, respectively. The loss of soluble solid content was the highest in the control group (3.78%), while the loss was 2.87% and 2.64% after CH and 2% CH/LS coating treatments. On the 6th day of storage, the soluble solids in the control group decreased rapidly. The reason may be that PE film group had poor water permeability, accumulated more water on the surface, and had a large number of microbial breeding [41], which accelerated the spoilage and deterioration of grapes. Therefore, the content of PE film group decreased steadily in the early stage, and rapidly decreased to 2.1% in the later stage. The soluble solid content of CH coating group is slightly higher than that of 2% CH/LS packaging, which may be due to that the 2% CH/LS composite film has good water retention and bacteriostasis, which can not only reduce the water loss of grapes but also inhibit the reproduction of spoilage bacteria [42]. However, the CH film has a large water vapor transmission coefficient and a fast water loss rate, leading to a slightly higher soluble solid content [43]. It indicated that CH coating and 2% CH/LS coating could effectively reduce the loss of soluble solids in grapes.

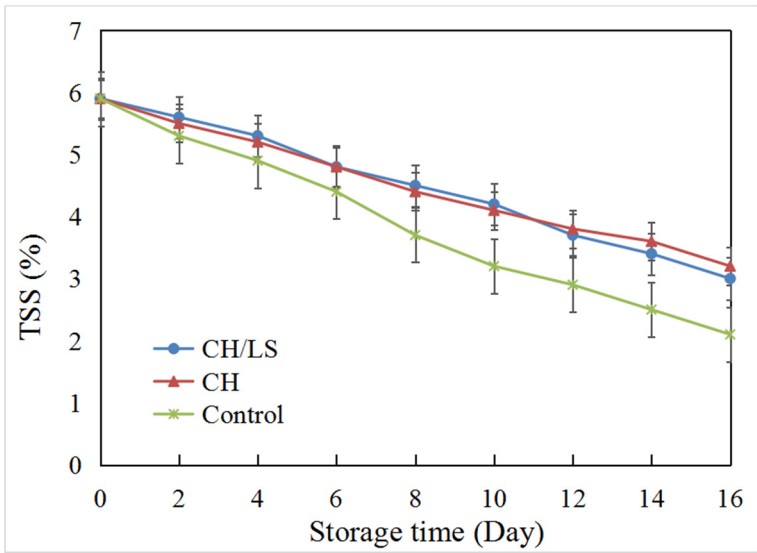

**Figure 4.** The Soluble solids of grapes during preservation with different packaging films.

### 3.1.5. Titratable Acid Content

The content of titratable acid is related to the color, aroma, taste and stability of the fruit [44]. With the prolongation of storage time, the titratable acid content of fruits showed a downward trend. Generally speaking, grapes with lower maturity will continue their physiological activities after picking, and the titratable acid may rise first and then fall [45]. However, mature grapes were selected in this study, so the titratable acid showed a continuous downward trend [46]. Figure 5 shows that the reduction rate of titratable acid of grapes coated with 2% CH/LS decreased slowly than the other two groups; during the 16th day of the experimental period, the titratable acid content decreased by only 1.39, and its content was 3.32 g/kg. In comparison, the control group decreased by 2.3, and its content was only 2.41 g/kg. CH and 2% CH/LS film-coated packaging groups decreased significantly lower than the control group ($p < 0.05$).

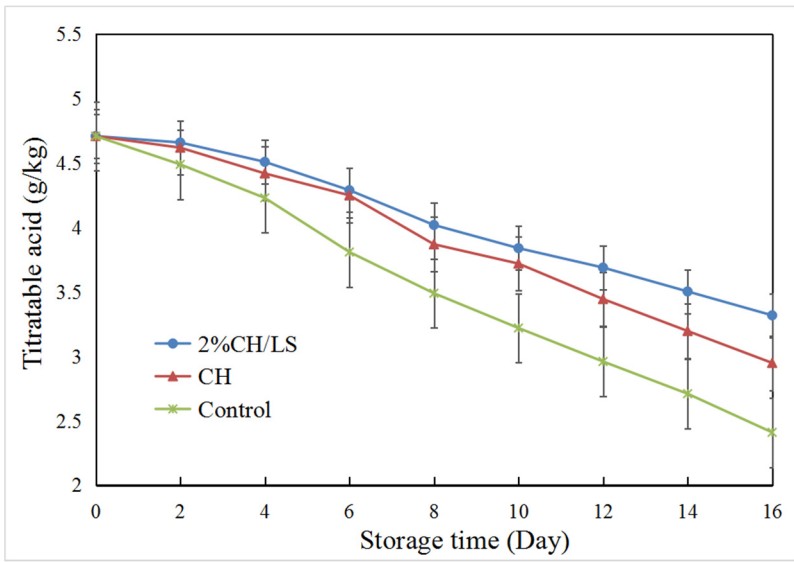

**Figure 5.** Titratable acid of grapes during preservation with different packaging films.

### 3.2. Identification of Wine Metabolites from Helan Mountain in Ningxia

Figure 6 shows the [1]H-NMR spectra of Cabernet Sauvignon dry red wine, Chardonnay and Italian Riesling dry white wine of Helan Mountain in Ningxia. The metabolite

displacement information in wine is combined with Figure 6 and references the relevant literature [47,48]. The results are shown in Table 1.

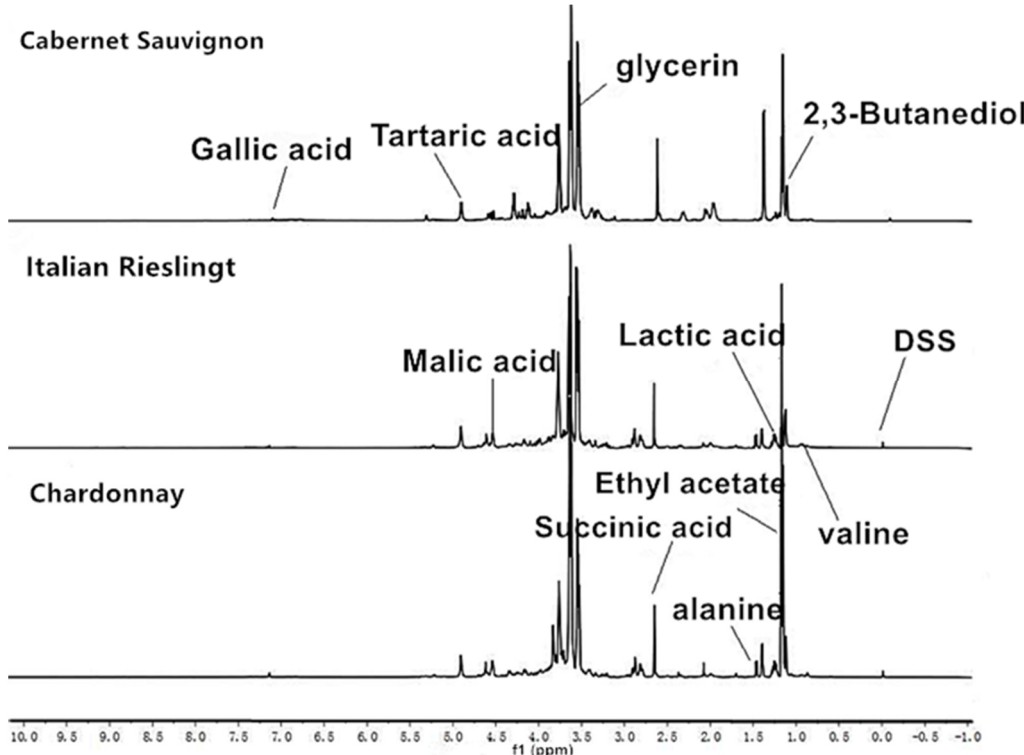

**Figure 6.** [1]H-NMR spectra of wine metabolites from Helan Mountain in Ningxia.

**Table 1.** [1]H-NMR assignment of metabolites in wines.

| Keys | Metabolites | [1]H-NMR Chemical Shift |
|:---:|:---:|:---:|
| 1 | Valine | 0.87 (d, $C4H_3$), 1.03 (d, $C5H_3$) |
| 2 | Ethanol | 1.18 (t, $C2H_3$), 3.64 (q, $C1H_2$) |
| 3 | 2,3-Butanediol | 1.16 (d, $C1H_3 + C4H_3$) |
| 4 | Succinic acid | 2.64 (s, $C2H_2 + C3H_2$) |
| 5 | Proline | 2.00 (m, u, $\gamma$-$CH_2$), 2.07 (m, u, $\beta$-CH), 2.35 (m, u, $\beta'$-CH), 3.35 (m, u, $\delta$-CH), 3.42 (m, u, $\delta$-CH), 4.16 (m, u, $\alpha$-CH) |
| 6 | Ethyl acetate | 1.26 (t, $C4H_3$), 4.16 (q, $C3H_2$) |
| 7 | Tartaric acid | 4.53 (s, C2H + C3H) |
| 8 | $\alpha$-Glucose | 5.23 (d, $\alpha$C1H) |
| 9 | $\beta$-Glucose | 4.61 (d, $\beta$C1H) |
| 10 | Lactic acid | 1.36 (d, $C3H_3$), 4.28 (m, C2H) |
| 11 | Gallic acid | 7.14 (s, C2H + C6H) |
| 12 | Glycerol | 3.56 (q, $C2H_2$), 3.65 (q, $C3H_2$), 3.81 (m, C1H) |
| 13 | $\alpha$-D-Glucuronic acid | 5.34 (d, C1H) |
| 14 | $\gamma$-Aminobutyric acid | 2.50 (t, $\alpha$-$CH_2$), 1.96 (m, $\beta$-$CH_2$), 3.05 (t, $\gamma$-$CH_2$) |
| 15 | Malic acid | 2.73 (dd, $\beta CH_2$), 2.86 (dd, $\beta'CH_2$), 4.46 (q, CH) |
| 16 | Alanine | 1.51 (d, $\beta CH_3$) |
| 17 | D-Sucrose | 5.43 (d, C1H), 3.55 (dd, C2H), 3.72 (dd, C3H), 3.90 (dd, C4H), 4.215 (d, C1'H), 4.05 (dd, C2'H), 3.88 (dd, C3'H) |

The characters in brackets refer to peak information: s, singlet; d, doublet; t, triplet; q, quartet; dd, doublet of doublets; m, multiple.

Because [1]H-NMR detection hardly needs sample pre-treatment, the inherent properties of the sample are well preserved. Using pattern recognition analysis combined with the chemical shift of nuclear magnetic spectrum, the characteristic variables with a large contribution to the difference between samples can be obtained, so as to identify the

metabolites causing the difference between samples. The metabolites in the $^1$H-NMR spectra of wine were identified, and these substances mainly included amino acids, organic acids, sugars, phenols and so on. It can be found that the composition of metabolites in these three wines is basically the same, which means that the composition of metabolites in wines is relatively stable, but there are differences in the content of metabolites between different types of wines. Each wine has its own fingerprint, and different types of wine metabolic profiles describe their physiological and biochemical states, which need to be processed to find their markers [49]. The positions of the characteristic peaks in the NMR spectra correspond to different types of metabolites in wine, and the peak intensity (such as area) represents the relative content of the corresponding metabolites.

*3.3. Differences of Metabolites in Different Wine Varieties from Helan Mountain*

In this experiment, the metabolite map data of three wines in the Helan Mountain production area of Ningxia were compared with the PLS-DA model to determine the main metabolites causing the difference between wine varieties. Firstly, Chardonnay and Italian Riesling dry white wine were analyzed by PLS-DA. Then, the pair PLS-DA comparison of dry white and dry red wine was carried out to determine the metabolites causing the difference between dry red and dry white wine varieties.

3.3.1. Analysis of Metabolites of Chardonnay and Italian Riesling Dry White

Two pairs of PLS-DA of dry white wine were compared to determine the main metabolites causing the difference between dry white wine varieties. The PLS-DA model of the 2020 Chardonnay and Italian Riesling Dry white wine is shown in Figure 7. In the score graph, Chardonnay and Italian Riesling dry white wine are clearly distinguished on the PC1 axis, and the cumulative contribution rate is $R^2X = 0.965$, $R^2Y = 0.994$ and $Q^2 = 0.955$, indicating that this model is effective. The validation diagram of the model in the permutation experiment further demonstrates the reliability and predictability of the model. It can be seen from the load diagram that compared with Chardonnay dry white wine, the 2,3-butanediol, lactic acid, succinic acid, glycerin, choline, tartaric acid, D-sucrose, and γ-aminobutyric acid content is relatively high in Italian Riesling Dry white wine, while the content of gallic acid, ethyl acetate, proline, malic acid, alanine, α-glucose and β-gluconic acid is relatively low.

In order to ensure the unique taste of dry white wine, the fermentation process of malic acid and lactic acid is properly controlled during the brewing of dry white wine. Malic acid plays an important physiological role in the human body. It can effectively improve the body's exercise ability, resist fatigue, accelerate the metabolism of carboxylate, protect the heart, improve memory, etc. [50]. From the perspective of organic acids, it can be considered that dry white wine has a relatively high protective effect on the body.

3.3.2. Analysis of Differences of Metabolites between Cabernet Sauvignon Dry Red Wine and Italian Riesling, Chardonnay Dry White Wine

Pairwise comparison PLS-DA were compared between dry white and dry red wines to determine the metabolites that caused the difference between the two wine varieties. Figure 8 shows the PLS-DA model of Cabernet Sauvignon dry red wine and Italian Riesling Dry white wine in 2020. In the score chart, the two wines are clearly distinguished on the PC1 axis, where the cumulative contribution rate is $R^2X = 0.76$, $R^2Y = 0.987$ and $Q^2 = 0.969$, indicating that the quality of this model is good. The validation diagram of the permutation experiment of this model once again shows the reliability and predictability of this model. It can be seen from the load diagram that compared with Cabernet Sauvignon dry white wine, Italian Riesling dry white wine has higher alanine and malic acid content, while 2,3-butanediol, glycerol, choline, lactic acid, valine, proline, ethyl acetate, succinic acid, tartaric acid, gallic acid, α-D-Glucuronic acid is low.

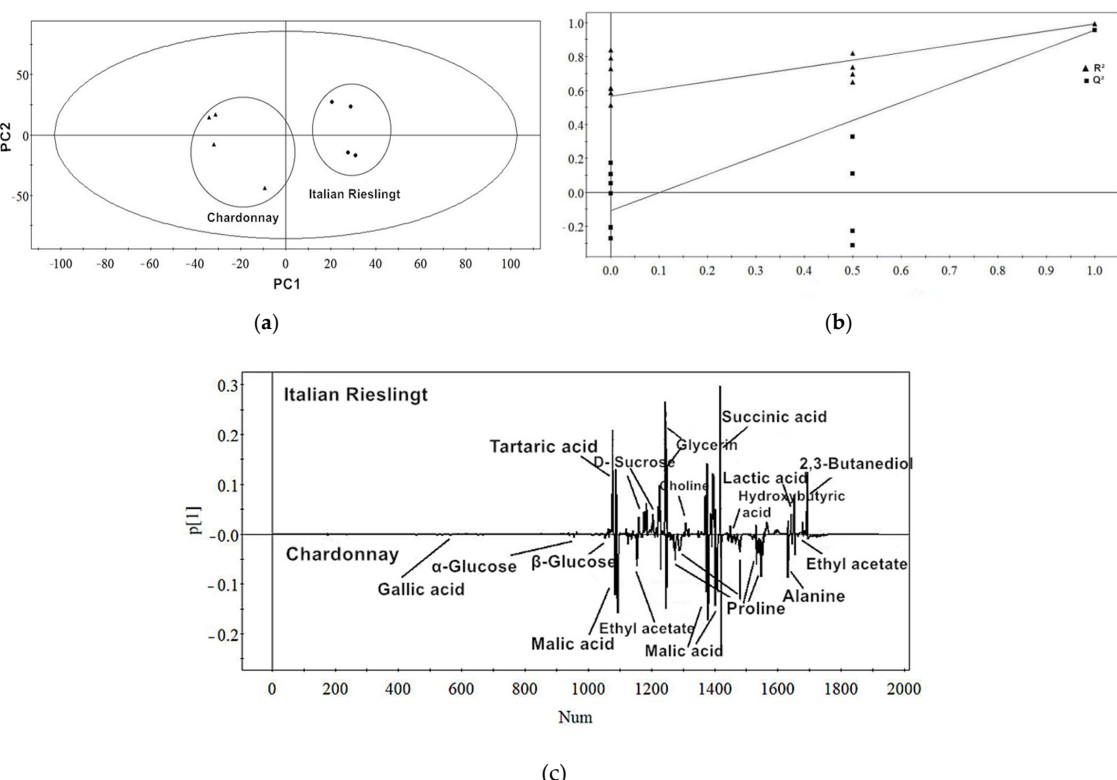

(c)

**Figure 7.** (**a**) PLS-DA model derived from the $^1$H-NMR spectra of Chardonnay and Italian Riesling dry white wine. PLS-DA scores plot; (**b**) PLS-DA cross-validation plot; (**c**) PLS-DA loading plot.

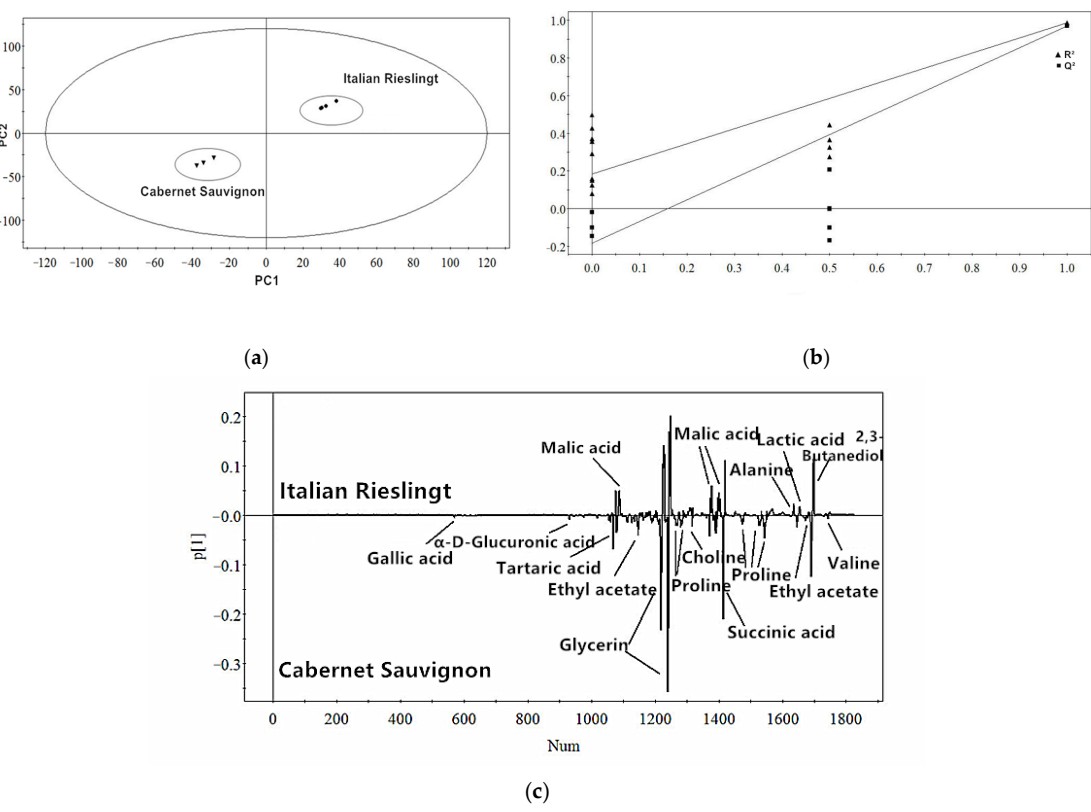

(c)

**Figure 8.** (**a**) PLS-DA model derived from the $^1$H-NMR spectra of Cabernet Sauvignon dry red wine and Italian Riesling dry white wine. PLS-DA scores plot; (**b**) PLS-DA cross-validation plot; (**c**) PLS-DA loading plot.

The metabolites of amino acid characteristic differences among wines detected in this experiment are valine, alanine and proline. Among them, valine and alanine contribute less to the difference between wines, while proline contributes more to the difference between different varieties of wines. Song et al. [51] also recognized that the content of proline in wine is affected by environmental factors and different varieties of wine grape berries.

The PLS-DA model of 2020 Cabernet Sauvignon and Chardonnay is shown in Figure 9. In the score graph, the two wines are significantly different on the PC1 axis, where the cumulative contribution rate $R^2X = 0.798$, $R^2Y = 0.987$ and $Q^2 = 0.979$ are relatively high, which also indicates that the model established is effective. The validation diagram of the permutation experiment further demonstrates the reliability and predictability of the model. As can be seen from the load diagram, Chardonnay dry white wine compared with Cabernet sauvignon dry red wine, Chardonnay dry white wine of alanine, malic acid content is higher, and 2,3-butanediol, valine, choline, glycerin, tartaric acid, lactic acid, valine, proline, ethyl acetate, succinic acid, gallic acid, $\alpha$-D-Glucuronic acid content is low.

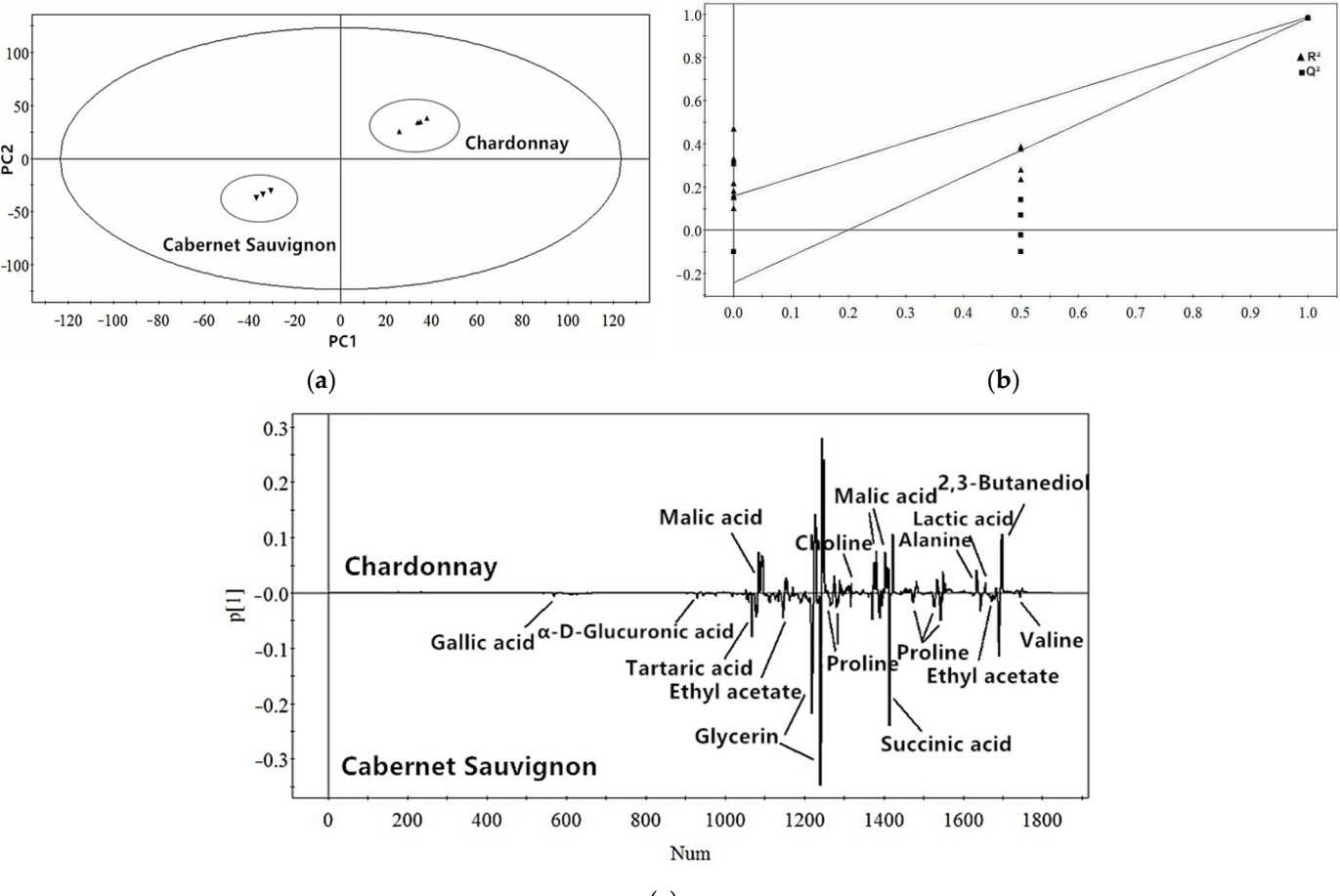

**Figure 9.** (**a**) PLS-DA model derived from the $^1$H-NMR spectra of Cabernet Sauvignon dry red wine and Chardonnay dry white wine. PLS-DA scores plot; (**b**) PLS-DA cross-validation plot; (**c**) PLS-DA loading plot.

As an important flavor substance of wine, organic acids not only determine the quality of wine [52], but also regulate the acid-base balance in the body, enabling the physiological activities of enzymes to be realized [53]. In this experiment, the different metabolites of organic acids were tartaric acid, malic acid, succinic acid and lactic acid. Tartaric acid and malic acid are derived from grape berries, while lactic acid and succinic acid are derived from wine fermentation [54].

Based on PLS-DA analysis of Cabernet Sauvignon, Chardonnay and Italian Riesling of the Helan Mountain region of Ningxia in 2020, it was found that the components of metabolites of different varieties of wine had little difference, but the content of metabolites had great difference. The contents of 2,3-butanediol, ethyl acetate, proline, succinic acid, tartaric acid, lactic acid, glycerin, gallic acid, choline, valine and α-D-Glucuronic acid in Cabernet Sauvignon dry red wine were higher than dry white wine. Ethyl acetate is an aromatic substance abundant in wine, which brings rich aroma to wine. Its content is also related to fermentation technology, grape varieties, fermentation temperature, etc. [55]. Ethyl acetate for wine varieties in this experiment and region identification provides a larger contribution, three different varieties of wine, the ethyl acetate content of Cabernet sauvignon is highest. The contents of malic acid, alanine and γ-aminobutyric acid in Chardonnay and Italian Riesling Dry white wine are higher than Cabernet Sauvignon dry red wine, and the contents of other nutritional metabolites are relatively low, indicating that there are significant differences in nutritional metabolites among different grape varieties. Thus, a quantitative analysis of the main metabolites was carried out, as shown in Table 2. The content of metabolites can be converted by the ratio of the peak area caused by protons on a specified group of the substance to be measured in the $^1$H-NMR spectrum to the peak area caused by protons on the specified group of the added internal standard DSS. The results were consistent with the results of PLS-DA analysis by comparing the contents of major metabolites by means of multiple samples (SNK method). The highest content of gallic acid was found in Cabernet Sauvignon. The content of alanine and malic acid is the highest in Chardonnay wine, which makes the wine made with different styles, guiding consumers to choose the appropriate wine according to their own needs and improving their health.

**Table 2.** Content of main metabolites in Cabernet Sauvignon dry red wine, Chardonnay, Italian Riesling dry white wine (g/L).

| Metabolites | Cabernet Sauvignon | Chardonnay | Italian Riesling |
|---|---|---|---|
| Ethyl acetate | 1.52 ± 0.03 b | 0.92 ± 0.01 c | 0.32 ± 0.01 d |
| Lactic acid | 0.51 ± 0.03 b | 0.22 ± 0.02 e | 0.32 ± 0.02 d |
| Alanine | 0.04 ± 0.01 d | 0.25 ± 0.02 a | 0.14 ± 0.01 b |
| Succinic acid | 1.31 ± 0.02 b | 0.46 ± 0.02 d | 0.47 ± 0.01 d |
| Proline | 3.23 ± 0.05 c | 1.50 ± 0.08 d | 0.49 ± 0.02 e |
| Malic acid | 4.95 ± 0.17 c | 6.08 ± 0.28 a | 5.37 ± 0.02 b |
| Choline | 0.06 ± 0.01 a | 0.04 ± 0.01 b | 0.05 ± 0.01 b |
| Glycerol | 13.99 ± 0.09 b | 8.92 ± 0.16 e | 9.89 ± 0.19 d |
| Gallic acid | 0.14 ± 0.01 a | 0.03 ± 0.00 c | 0.01 ± 0.00 c |

Means followed with different letters are statistically different at the 0.05 probability level with an AVOVA-protected SNK 0.05 test.

## 4. Conclusions

In this study, lignosulfonate (an underutilized renewable biomass resource) was added to chitosan solution, and a separate 2% chitosan film (CH), chitosan—1% sodium lignosulfonate film (1% CH/LS) and chitosan—2% sodium lignosulfonate film (2% CH/LS) biomass composite films were developed by the classical casting method, to further study the preservation effect of 2% CH/LS film and CH film on grapes. From the test results, it can be concluded that 2% CH/LS films showed the best preservation performance. Compared with the control, CH film and 2% CH/LS coating film packaging not only effectively inhibit the evaporation of water and slow down the weight loss rate of grapes during storage, effectively alleviating the hardness, soluble solids, and titratable acid of grapes and causing the decrease of other nutrients, but they also effectively extend the shelf life of grapes. Therefore, compared with ordinary plastic packaging, 2% CH/LS film packaging is one of the promising strategies for preservation, as well as the Chitosan-lignosulfonate composite coating film. Both will be further investigated for various fruit and vegetable preservation. At the same time, this paper, based on the method of $^1$H-NMR combined with pattern



recognition analysis, analyzed the differences in metabolites in Cabernet Sauvignon dry red wine, Chardonnay and Italian Riesling dry white wine of Helan Mountain in Ningxia. As far as the research results are concerned, the types of metabolites of the three wines are similar, but their content is significantly different. Among them, Chardonnay wine has a more refreshing and delicate taste and more complete flavor, while Cabernet Sauvignon wine has the highest biological activity and health function. This provides theoretical and technical support for the quality control and evaluation of the origin of wine, the identification and protection of varieties, and provides a scientific guide for consumers to aid in the prevention of diseases and the maintenance of human health.

**Author Contributions:** Writing—original draft preparation, L.L.; writing—review and editing, B.H. and X.Z.; designed research, Y.F. and L.L.; validation, M.Z.; analyzed data, M.Z. and L.L.; supervision, X.Z. All authors have read and agreed to the published version of the manuscript.

**Funding:** This research was funded by National Natural Science Foundation of China "Wine Metabolomics and NMR Fingerprint Stud, grant number NO. 31271857".

**Institutional Review Board Statement:** Not applicable.

**Informed Consent Statement:** Not applicable.

**Data Availability Statement:** All raw data are available at the corresponding author.

**Acknowledgments:** We acknowledge financial support by the National Natural Science Foundation of China and President Zhou's Laboratory for its help.

**Conflicts of Interest:** The authors declare no conflict of interest.

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
