# Peer review of "Application of Chitosan-Lignosulfonate Composite Coating Film in Grape Preservation and Study on the Difference in Metabolites in Fruit Wine"

_coatings, doi:10.3390/coatings12040494_

Round 1

Reviewer 1 Report

The work is interesting, the methods are good explained. I think that it wolud be good to put results and discussion together, not seperated. The comments are in the manuscript. 

Author Response

Thank you for your suggestions. Results and discussion have been combined, Please see the attachment.

Reviewer 2 Report

They are some problems with this manuscript. No attention is paid to precision in descripting methods or presenting results, even they are enough and the methods used are adequate. The English language is very strange; this is the major aspect to be improved here.

Detailed observations:

A literature source should be given for the affirmations at page 2, lines 86-88. No discussion on PLS-DA is given in reference [25].

In Material and methods, it is not clear explained what means “rotted” (equation 1). They should verify if they used correctly the term “Rotation rate”, I thing they wanted to write “rot index”, as presented in Results. Anyway, they should decide what they measure: rotation rate, rot index or decay rate, they are all in text, in different places.

Not clear what “appropriate amount of samples” at line 149 means.

Reagents and equipment are simply enumerated, there is no verb.

Figure 1: attention at the measure unit for the storage time, it is not given.

Figure 2: firmness or hardness is measured here?

Authors should pay more attention on the aspect of the figures.

More references should be included in Discussion.

English language has to be improved in a lot of places. Examples:

  • Page 2, lines 51-52: “in the fruit surface can form a layer of colorless transparent semi-permeable membrane”
  • Page 2, line 53: “fruit to reduce the water loss rate of the fruit”
  • The word “fruit” is repeated 3-4 times in the same proposition
  • Page 2, lines 56-57: “Long drying time, high water permeability and 56 poor toughness of the coating film”
  • Page 2, line 64: “Wine is a fermented wine”
  • Page 2, line 97: “This experiment explored”
  • Page 3, first paragraph has to be totally written, it is not understandable
  • Lines 111-112: combine substantive at singular with verb at plural
  • By describing the method for measuring firminess, titratable acidity or pretreatment of wine samples, the authors have to decide which verbal time they use. Normally, the past tense is used to describe a finished action
  • Line 140; “groups fruits”
  • ??? at line 215: “After CH and 2%CH/LS coating film packaging, a certain fresh-keeping effect was achieved, and the rot 216 of grapes is effectively slowed down.

And so on…  many wrong formulations.

Author Response

Thank you for your criticism and correction. Reference 25 has been corrected due to my negligence in introducing mistakes. Reagents and equipment have been linked into sentences with verbs. The mistakes you pointed out in English wording and grammar have been corrected. Please see the attachment.

Reviewer 3 Report

The manuscript entitled: "Application of Chitosan-lignosulfonate composite coating film in grape preservation and study on the difference of metabolites in fruit wine" is about the preparation and application of a mixed biopolymer coating film for grape preservation. In general, the manuscript is interesting, well designed, and has enough novelty for publication. There are some comments to address by authors before final decision as follow:

Abstract: It is a long abstract; make it short and informative. Also, you need to support the results in the abstract with some quantitative data.

Keywords: Choose keywords other than the main words in the title. This will help the other researchers to find your article by searching a wide range of word searches through databases.

Introduction: The introduction part needs to improve in implementing the recent publication. For example, in the chitosan application or in terms of application of a coating to other fruits, you can improve by the implementation of related articles such as Esmaeili, Yasaman, et al. "The synergistic effects of aloe vera gel and modified atmosphere packaging on the quality of strawberry fruit." Journal of Food Processing and Preservation 45.12 (2021): e16003. At the end of the introduction, clearly bring your objectives without details.

Materials: The subheading of the material part should not come 2.1, not under other subheadings.

Statistical analysis: You need to add details of statistical analysis in both methodology and results.

Results: Figure 1 should be redrawn in line style like other your figures because the X-Axis is time and it is a continuous variable.

Table 2: Better to report the results with a maximum of 2 figures after ., eg: 1.52 instead of 1.519.

Conclusion: This part is too long. You must shorten this part, justify your hypothesis, and if you have any recommendations. No need to repeat the results.

Author Response

Thank you for your suggestions. The summary and conclusion parts have been briefly modified, and quantitative data have been added. Key words have been modified and added, statistics and analysis parts have been modified as suggested. Please see the attachment. However, I think FIG. 1, presented in columnar form, can more intuitively show the different effects of the three treatments on fruit decay rate and the trend of prolonged preservation time in the experimental group. Please see the attachment.

Round 2

Reviewer 3 Report

Unfortunately, the authors did not care about the comments. For example conclusion, you only changed two paragraphs with each other. Also about Figure 1, it is not my point to redraw the figure, it must be changed to have meaning. So, I do not recommend the manuscript for publication.

Author Response

Dear reviewer, first of all, thank you very much for your valuable comments and suggestions, After several discussions with other authors, we approve of change figure 1 into a broken line chart like other figures, and change the black line to color for readers to observe and distinguish. As for the conclusion part, we revised it again, did not repeat the result part and tried our best to make it brief. Please see the attachment for details.

Round 3

Reviewer 3 Report

Improved and acceptable for publication